# OpenReview forum: "Activation Oracles: Training and Evaluating LLMs as General-Purpose Activation Explainers"
_ICML.cc/2026/Conference — ICML 2026 spotlight_

### Official Review · Reviewer_6w6M · 2026-02-25

**Soundness:** 3
**Presentation:** 4
**Significance:** 3
**Originality:** 2
**Overall Recommendation:** 5
**Confidence:** 2

**Summary:**

The paper trains and evaluates LatentQA models, language models that answer questions about the activations of a target model. Different from prior work, this paper carries out the training and evaluation of LatentQA models in a broad setting, across multiple tasks and datasets. A main finding of the paper is that LatentQA models generalize on out-of-distribution tasks, especially when scaling up the number of training tasks and samples.

**Compliance With Llm Reviewing Policy:**

Affirmed.

**Final Justification:**

The authors gave satisfactory answers to my questions. I therefore maintain my positive score '5: Accept'.

- The paper is well-written.
- The contributions are significant as the paper provides evidence that LatentQA models can be a general-purpose technique to increase our understanding in model activations.
- The claims in the paper are soundly backed by experiments.
- A weaker point of the paper is that the delta with prior work is relatively small, both with respect to methods and new insights.

**Key Questions For Authors:**

1. Why use the different setup for Claude 3.5 Haiku? Especially, the choices to:
- Inject activations by replacing them instead of steering.
- Always inject activations from the same layer.
- Use different heuristics for selecting the token positions.

2. Did you do any analysis with respect to the layers from which you extract the target model activations?  Are there patterns in which layers work well for which method and/or datasets (or classification classes)? On PersonaQA, for instance, the three techniques have similar performance for Claude 3.5 Haiku (left side of Figure 9), which unlike the other methods only collects activations from a middle layer. Are these activations perhaps easier to explain for the baselines?

**Limitations:**

Yes.

**Strengths And Weaknesses:**

**Strengths**:
+ [Soundness] The claims of the paper are supported through experiments with language models of different sizes and on different tasks.
+ [Presentation] All methods and experiments are clearly explained, and the text is supported with useful illustrations.
+ [Presentation] The authors released the code and data (which seems to be self-contained and well-organized).
+ [Significance] The work provides evidence that LatentQA models can be a general-purpose technique to increase understanding in model activations, which can make the technique more impactful.

**Weaknesses**:
- [Soundness] A different training and evaluation setup is used for the Claude 3.5 Haiku (appendix B.6) activation oracle. The authors don't provide the reason(s) behind the differences. They seem quite ad hoc.
- [Significance] LatentQA models are themselves difficult to understand and therefore do not help our understanding of underlying model mechanics.
- [Originality] That LatentQA models can generalize on out-of-distribution tasks is an interesting insight, but beyond this the delta with prior work is rather small.

Minor comments:
- The authors' choice to brand general-purpose LatentQA models as Activation Oracles a bit unfortunate. Oracles are typically systems that (always) provide the correct answers.
- Reporting results that require whitebox access to a closed-weight model leaks information about the authors.

---

> ### Author Rebuttal · Authors · 2026-03-31
>
> We thank the reviewer for their thoughtful review.
>
> **Ad hoc Claude Haiku 3.5 setup (W1, Q1)**
>
> We acknowledge that the Claude setup is ad hoc. The Claude experiments were implemented separately from the open-source models, and the differences primarily arose from infrastructure constraints. We will clarify this in the camera-ready.
>
> That said, the core empirical claim does not rely on Claude, as the main finding that diverse training improves OOD transfer already holds across the three open-weight models under a uniform pipeline. We view Claude as supplementary evidence that some transfer extends to a more capable closed-weight model.
>
>
> **AOs do not help understanding of underlying model mechanics (W2)**
>
> We agree, and this is a trade-off that we discuss in Section 7 and Appendix A. AOs are fundamentally non-mechanistic: they cannot answer questions about how meaning is encoded or how computations transform activations. They are also not exhaustive (unlike SAEs, which can in principle surface meaning we don't know to look for), and their expressivity means they may assemble conclusions the target model has not itself represented. On the other hand, AOs can generalize to new questions without retraining a separate probe per property, and applying other white-box techniques to our auditing tasks requires task-specific scaffolding and tuning. We view AOs as complementary to mechanistic methods and will expand this discussion in the camera-ready.
>
> **Small delta with prior work (W3)**
>
> We agree that the algorithmic delta is small. Our contribution is primarily empirical: (1) training on more diverse data mixtures and (2) evaluating out-of-distribution generalization, especially on downstream auditing tasks. Prior to this work, it was not established that activation verbalization models could transfer to tasks qualitatively unlike their training data. We also provide controlled ablations showing that diversity, not just quantity, drives OOD transfer (Figures 6, 10).
>
> **Layer analysis (Q2)**
>
> For all evaluation tasks (Taboo, PersonaQA, Gender, SSC), we use activations from layer 50% depth. We examine the effect of layer selection in the appendix: Figure 14 shows that classification performance is consistent across trained layers (25%, 50%, 75%) and generalizes to untrained layers (33%, 66%), and Figure 15 shows that layer 50% outperforms layer 75% on the secret-keeping evaluations. So mid-layer activations are a strong default in our experiments.
>
> For PersonaQA, all models use activations from the same 50% depth layer, so the similar performance across methods for Claude may reflect differences in how Claude internalized the fine-tuned knowledge rather than a layer depth effect.

---

> > ### Author Rebuttal · Reviewer_6w6M · 2026-04-01
> >
> > My questions have been adequately addressed. As such, I maintain that '5. Accept' is an appropriate score.

---

### Official Review · Reviewer_cJWB · 2026-03-12

**Soundness:** 3
**Presentation:** 3
**Significance:** 3
**Originality:** 2
**Overall Recommendation:** 4
**Confidence:** 4

**Summary:**

The paper investigates fine-tuning models to answer questions about another model's internal representations.  For example answering questions about a sysem prompt, or secret that the model was not trained to generate.

**Compliance With Llm Reviewing Policy:**

Affirmed.

**Final Justification:**

I think that although the paper has a few limitations, this is nice work.

**Key Questions For Authors:**

I am curious what the authors see could be a practical use-case for the proposed method?

**Limitations:**

Yes

**Strengths And Weaknesses:**

Strengths:
- Training models to verbally interpret hidden activations is an interesting and under-explored direction.
- The proposed method seems sensible.
- The paper is written clearly.

Weaknesses
- The main results in Figure 3 are a bit weak relative to a simple prompting approach (e.g., "The secret is:") for Gender and Secret Side Constraint.
- The paper doesn't offer a lot in terms of novel insights or other contributions relative to LatentQA and Cywinski et. al. (both of which are cited).

---

> ### Author Rebuttal · Authors · 2026-03-31
>
> We thank the reviewer for their thoughtful review.
>
> **Performance on Gender and SSC (W1)**
>
> We agree that AO performance can be weak relative to the best black-box baselines, especially on the SSC evaluation. We note that on Gender, the gap is small and within error bars (97.0% for AO vs 97.7% for the best black-box baseline). However, AOs outperform all white-box baselines on all three secret-keeping tasks, and on Taboo they also outperform all black-box baselines (21% vs 4-7%).
>
> **Novelty relative to LatentQA and Cywinski et al. (W2)**
>
> We do not claim large algorithmic novelty over LatentQA and view our contribution as primarily empirical: (1) training on more diverse data mixtures and (2) evaluating out-of-distribution generalization, especially on downstream auditing tasks. Prior LatentQA work (Pan et al., 2024) was evaluated only on narrow, in-distribution tasks, such as extracting system prompt information or answering simple questions about the activations of entities (e.g. "Which sport does this person play?", using the activation for "Michael Jordan"). Our central finding is that these methods generalize far out of distribution: AOs trained on system prompt QA and simple classification tasks can recover information that was fine-tuned into a model but never appears in the input text. We also provide systematic ablations (Figures 6, 10) showing that improved training data diversity is an important driver of this generalization.
>
> **Practical use-cases (Q1)**
>
> We view AOs broadly as a more accessible form of interpretability, providing a natural-language interface for asking questions about model behavior without requiring expertise in specialized tools such as SAEs, logit lens, or circuit analysis. Any setting where practitioners want to understand what a model is doing internally is a potential application.
>
> One concrete example is proposing hypotheses about the effects of a fine-tuning or post-training run: given a fine-tuned model, an AO can quickly produce natural-language hypotheses about what changed or hidden objectives that may be present. This can guide targeted red-teaming, evaluation design, or further mechanistic analysis. Our model diffing (Section 4.2) and hidden-objective screening (Section 4.1) evaluations demonstrate this kind of application, albeit in controlled benchmark settings.

---

> > ### Author Rebuttal · Reviewer_cJWB · 2026-04-01
> >
> > Thanks for the response!  I think that although the paper has a few limitations, this is nice work.  Perhaps it would help to add some discussion on potential practical use cases and future work.

---

### Official Review · Reviewer_uDdR · 2026-03-13

**Soundness:** 4
**Presentation:** 3
**Significance:** 3
**Originality:** 3
**Overall Recommendation:** 5
**Confidence:** 4

**Summary:**

This paper extends the LatentQA-style activation question-answering approach into a more general framework called Activation Oracles (AOs): training a model to take internal activations from a target LLM as an input modality and answer natural language questions about these activations. The authors train AOs using a richer mixture of training tasks and evaluate their generalization on highly out-of-distribution auditing tasks, including secret-keeping scenarios and emergent misalignment model-diffing audits. Experiments show that AOs match or exceed existing white-box and black-box baselines across multiple tasks, with both the quantity and diversity of training data significantly impacting downstream generalization performance.

**Compliance With Llm Reviewing Policy:**

Affirmed.

**Final Justification:**

The rebuttal has resolved my issue. I raised my score to 5.

**Key Questions For Authors:**

1. AO output error rates and confidence calibration are critical for actual auditing. Have you tried uncertainty expression/abstention training?
2. Current evaluations mostly involve narrow fine-tuning behaviors. Do you have evidence that AOs can effectively explain more realistic post-training changes? Preliminary results would significantly strengthen extrapolation and impact.

**Limitations:**

yes

**Strengths And Weaknesses:**

Strengths
- The problem definition is clear, experimental objectives are well-defined and the core claim is supported by multi-task, multi-model experiments. The training data mixture and ablation design are convincing, especially the distinction between quantity and diversity which adds value.
- This work addresses an important problem that still lacks a unified interface and proposes a practical solution. If the method proves robust, it could lower the barrier for non-mechanistic-interpretability experts to use activation explanations, offering real value for safety auditing toolchains.
- While the core components aren't entirely new, the paper's contribution lies in systematizing them into a general Activation Oracle paradigm and demonstrating generalization to safety auditing tasks quite different from the training distribution.
- The overall narrative is fairly complete, with coherent logic connecting training tasks, evaluation tasks, and ablations. Figures help illustrate the differences between training tasks and OOD evaluation tasks.

Weaknesses
- AO outputs are natural language guesses/explanations, and the paper acknowledges that models often guess incorrectly and don't calibrate uncertainty well. This limits their reliability in real safety auditing scenarios. Current results demonstrate more potential usability rather than highly reliable auditing tools.
- Evaluation tasks are mainly based on relatively narrow fine-tuning behaviors. While reasonable for method validation, extrapolation to real-world complex post-training remains unclear.
- The relationship with existing PatchScopes/SAE/LatentQA/Meta-Models and other internal state textualization approaches is clear, but the trade-off discussion between "usability vs verifiability" with verifiable mechanistic interpretation could be deeper.

---

> ### Author Rebuttal · Authors · 2026-03-31
>
> We thank the reviewer for their thoughtful review.
>
> **Reliability, calibration, and uncertainty expression (W1, Q1)**
>
> We agree that uncertainty handling is critical for practical deployment. Our current evidence is preliminary, but we have found two promising directions:
>
> 1. **Binary tasks:** Using the logit difference between "yes" and "no" tokens as a confidence score enables a precision-recall trade-off at zero additional cost. On the same OOD classification setting used in the main paper with Qwen3-8B, ranking by absolute logit margin yields PR AUC = 0.754, and restricting to the 10% most confident predictions increases accuracy from 70.1% at full coverage to 87.2%.
> 2. **Open-ended tasks:** Sampling 10 rollouts at temperature 1 and computing majority-vote consensus provides a useful proxy for correctness. On Taboo with Qwen3-8B, this yields PR AUC = 0.874, and restricting to predictions with consensus >= 0.9 (~13% coverage) increases accuracy from 46.6% to 96.5%.
>
> Both approaches operate purely at inference time and require no architectural changes. The consensus approach could also serve as a training signal for future work. We will include these results in the camera-ready version, if accepted.
>
> **Narrow fine-tuning benchmarks and realistic post-training (W2, Q2)**
>
> We agree that evaluation on more realistic post-training settings would be valuable. We note that the benchmarks we use (from Cywinski et al., 2025 and Minder et al., 2025) are among the most established in this area, and we believe developing improved benchmarks is important future work.
>
> As additional preliminary evidence of broader applicability, we have results applying AOs to non-fine-tuned model behaviors. These do not directly address realistic post-training pipelines, but suggest AOs can extract meaningful signals beyond our current narrow fine-tuning benchmarks.
>
> These results use our existing AO with no task-specific training, purely testing generalization from our current recipe. AOs can detect sycophantic behavior, distinguishing cases where a model flips its answer to agree with a user from cases where it was already going to answer that way (AUC = 0.83). AOs can also predict whether a model will answer MMLU questions correctly by reading off internal confidence (AUC = 0.75). We expect these results could improve further with targeted training data.
>
>
> **Usability vs. verifiability trade-off (W3)**
>
> We agree that this trade-off deserves deeper discussion. AOs are fundamentally non-mechanistic: they cannot answer questions about how meaning is encoded or how computations transform activations. They are also not exhaustive (unlike SAEs, which can in principle surface meaning we don't know to look for), and their expressivity means they may assemble conclusions the target model has not itself represented. On the other hand, AOs can generalize to new questions without retraining a separate probe per property, and applying other white-box techniques to our auditing tasks requires task-specific scaffolding and tuning. We view AOs as complementary to mechanistic methods and will expand this discussion in the camera-ready version.

---

> > ### Author Rebuttal · Reviewer_uDdR · 2026-04-04
> >
> > Thanks for your rebuttal. I have no other questions.

---

### Official Review · Reviewer_y2pN · 2026-03-15

**Soundness:** 4
**Presentation:** 3
**Significance:** 4
**Originality:** 3
**Overall Recommendation:** 5
**Confidence:** 4

**Summary:**

This paper studies Activation Oracles (AOs): LLMs fine-tuned (via LoRA) to accept intermediate activation vectors from a target LLM as input and answer arbitrary natural-language questions about them. Building on the LatentQA framework of Pan et al. (2024), the authors scale training data diversity by combining three dataset types: system prompt question-answering (SPQA), binary classification tasks (7 datasets), and a self-supervised context prediction task (predicting tokens before/after a given activation sequence). AOs are evaluated on four substantially out-of-distribution downstream auditing tasks from recent safety-oriented work: recovering secret words from Taboo-playing models, detecting user gender from fine-tuned models, extracting base64-encoded side constraints, and auditing emergent misalignment via model diffing. The authors demonstrate that AOs generalize far beyond their training distribution, matching or exceeding all white-box interpretability baselines on all four tasks and the best overall baseline on 3 of 4. Ablations across four models (Qwen3-8B, Gemma-2-9B-IT, Llama-3.3-70B-Instruct, Claude Haiku 3.5) show that scaling the diversity of training data consistently improves OOD generalization.

**Compliance With Llm Reviewing Policy:**

Affirmed.

**Key Questions For Authors:**

How do you envision calibrating AO outputs for safety-critical applications? Could you train AOs to express uncertainty (e.g., via verbalized confidence or abstention), and would this be compatible with the current activation steering approach?

**Limitations:**

Yes, the limitations are discussed honestly in Section 7 and Appendix A, including the lack of calibration, narrow fine-tuning benchmarks, and the black-box nature of AOs (sacrificing mechanistic understanding for a question-answer interface). The paper would benefit from a more explicit discussion of the gap between controlled benchmarks and realistic post-training pipelines.

**Strengths And Weaknesses:**

Strengths:
1. The practical implications for AI safety auditing are substantial. Providing a natural-language interface to LLM activations could democratize interpretability, making it accessible to practitioners who lack expertise in SAE feature analysis, logit lens, or circuit-level methods. The ability to recover information that was fine-tuned into a model but never appears in the input text (e.g., Taboo secret words, PersonaQA attributes) is a striking demonstration of out-of-distribution generalization.

2. Soundness: The evaluation is commendably thorough: four target models (including a closed-weight Claude model), four downstream auditing tasks with established baselines from Cywinski et al. (2025) and Minder et al. (2025), systematic ablations across training mixtures (Figure 6, showing 10/12 model-task combinations peak with full mixture), and controlled experiments separating data diversity from quantity (Figure 10 / Appendix D). The learning rate sweep (Figure 7) shows the full mixture is more robust to hyperparameter choices than SPQA-only training.

Weaknesses:

1. Absolute performance levels remain concerning. On Taboo, the best AO achieves 20.5% accuracy—substantially better than the 4.4% white-box baseline, but still meaning ~80% of secrets are not recovered. On PersonaQA (open-ended), open-source AOs reach only ~14%. It is difficult to rule out that the successful cases reflect exploitable statistical regularities (e.g., certain secret words being more common or certain activation patterns being easier to read) rather than genuine general-purpose activation reading. The paper does not include adversarial controls (e.g., injecting random or shuffled activations) to establish a noise floor.

---

> ### Author Rebuttal · Authors · 2026-03-31
>
> We thank the reviewer for their thoughtful review.
>
> **On absolute performance levels:** We agree that absolute scores leave room for improvement. However, the 4.7× improvement over the best white-box baseline (20.5% vs. 4.4% on Taboo) suggests AOs are extracting meaningful signal that other methods miss.
>
> We also note that the ground-truth ceiling for how much information is recoverable from activations is generally unknown. While not directly comparable to our setting, Li et al. (2025b) found that even linear probes trained with full supervision achieved only ~25% accuracy on a related PersonaQA task, suggesting the ceiling for recoverable information from activations may be inherently low.
>
> **On adversarial controls / noise floor:** We agree that shuffled/random-activation controls would be a useful baseline for future work to establish a tighter noise floor.
>
> **Calibrating AO outputs / uncertainty expression:** We have preliminary but encouraging results:
>
> 1. **Binary tasks:** Using the logit difference between "yes" and "no" tokens as a confidence score enables a precision-recall trade-off at zero additional cost. On the same OOD classification setting used in the main paper with Qwen3-8B, ranking by absolute logit margin yields PR AUC = 0.754, and restricting to the 10% most confident predictions increases accuracy from 70.1% at full coverage to 87.2%.
> 2. **Open-ended tasks:** Sampling 10 rollouts at temperature 1 and computing majority-vote consensus provides a useful proxy for correctness. On Taboo with Qwen3-8B, this yields PR AUC = 0.874, and restricting to predictions with consensus >= 0.9 (~13% coverage) increases accuracy from 46.6% to 96.5%.
>
> Both approaches are fully compatible with the existing activation steering framework, as they operate purely at inference time and require no architectural changes. The consensus approach could also serve as a training signal for future work. We will add a discussion of these confidence strategies in the camera-ready version, if accepted.

---

> > ### Author Rebuttal · Reviewer_y2pN · 2026-04-06
> >
> > The authors agreed with the points raised by my reviews and I would just keep the score the same.

---

### Decision · Program_Chairs · 2026-04-30

**Decision:**

Accept (spotlight)

**Comment:**

This paper proposes Activation Oracles, a general-purpose framework that trains LLMs to interpret internal model activations and answer arbitrary natural language questions about them. The reviewers appreciated the work's thorough multi-model and multi-task evaluation, its clear presentation, and the practical implications for democratizing AI safety auditing without requiring mechanistic interpretability expertise. While some reviewers initially raised concerns regarding absolute performance levels, uncertainty calibration, and the incremental algorithmic novelty compared to prior work, the authors provided comprehensive responses. During the rebuttal period, they successfully addressed these issues by demonstrating confidence calibration strategies using logit margins and majority-vote consensus, alongside providing preliminary results on non-fine-tuned behaviors to validate broader applicability. Given the solid experimental design demonstrating strong out-of-distribution generalization and the authors' thorough addressing of reviewer concerns, I recommend accepting this paper for publication at ICML 2026.